# Textile Electrocardiogram (ECG) Electrodes for Wearable Health Monitoring

**DOI:** 10.3390/s20041013

**Published:** 2020-02-13

**Authors:** Katya Arquilla, Andrea K. Webb, Allison P. Anderson

**Affiliations:** 1Ann and H. J. Smead Department of Aerospace Engineering Sciences, University of Colorado Boulder, Boulder, CO 80303, USA; allison.p.anderson@colorado.edu; 2The Charles Stark Draper Laboratory, Inc., Cambridge, MA 02139, USA; awebb@draper.com

**Keywords:** smart textiles, e-textiles, wearables

## Abstract

Wearable health-monitoring systems should be comfortable, non-stigmatizing, and able to achieve high data quality. Smart textiles with electronic elements integrated directly into fabrics offer a way to embed sensors into clothing seamlessly to serve these purposes. In this work, we demonstrate the feasibility of electrocardiogram (ECG) monitoring with sewn textile electrodes instead of traditional gel electrodes in a 3-lead, chest-mounted configuration. The textile electrodes are sewn with silver-coated thread in an overlapping zig zag pattern into an inextensible fabric. Sensor validation included ECG monitoring and comfort surveys with human subjects, stretch testing, and wash cycling. The electrodes were tested with the BIOPAC MP160 ECG data acquisition module. Sensors were placed on 8 subjects (5 males and 3 females) with double-sided tape. To detect differences in R peak detectability between traditional and sewn sensors, effect size was set at 10% of a sample mean for heart rate (HR) and R-R interval. Paired student’s t-tests were run between adhesive and sewn electrode data for R-R interval and average HR, and a Wilcoxon signed-rank test was run for comfort. No statistically significant difference was found between the traditional and textile electrodes (R-R interval: t = 1.43, *p* > 0.1; HR: t = −0.70, *p* > 0.5; comfort: V = 15, *p* > 0.5).

## 1. Introduction

Health monitoring using wearable sensor systems is a rapidly expanding area of research that promises an increase in the availability of health data [1,2,3,4,5]. Cardiac health monitoring through electrocardiography (ECG) can be used to monitor heart-rate variability (HRV), which is an important metric for cardiac illness [6]. From the ECG signal, R-R interval (time between heartbeats) and heart rate (HR)—number of beats per minute (bpm)—can be derived. These measurements differ in that R-R interval gives a beat-to-beat variation metric, while HR is an average over the course of the monitoring period. ECG monitoring traditionally relies upon adhesive electrodes stuck on the skin for an extended period of time [7]. There are three key challenges that arise with this type of electrode: (1) adhesives can cause discomfort when left on the skin for an extended period of time; (2) the monitoring system is not well concealed and can be obvious to peers; and (3) the conductive gel within the electrodes can dry up over time, impacting the signal quality [8,9]. A potential solution to all three of these challenges is to fabricate “dry” electrodes that are directly integrated into clothing. This type of electrode does not have any conductive gel, so it will not dry out over time. These electrodes can be fabric-based or can utilize other substrates, such as flexible elastomers [10]. The potential advantage of dry electrodes that are textile integrated is that they are both flexible (making them more conformal to the body than traditional rigid disk electrodes) and washable, so it is feasible to use and reuse them. This reduces the consumables required to conduct long-term health monitoring, which is essential for applications in operational environments such as military and space operations. Additionally, integrating electrodes into textiles has the potential to increase comfort for the wearer, which in turn can increase the willingness to use the system and thus can increase the quantity of data collected. The development of “smart” textiles (textiles fabricated with conductive elements directly integrated) is on the rise, creating new opportunities for integrating advanced monitoring technologies into everyday items [11,12].

The development of flexible, dry electrodes has been explored through a variety of means, falling loosely into two categories: 1) conductive threads and yarns integrated into fabrics through traditional clothing fabrication methods and 2) conductive inks and pastes applied to fabrics through a variety of methods, such as stenciling, screen printing, and sputtering. Table 1 shows a broad range of groups working in this space with their chosen material and application method. Mestrovic et al. demonstrated the functionality of knitted textile electrodes for ECG monitoring using three different types of threads. They found success with these electrodes, but only tested them on one subject [13]. The yarn structure created through knitting consists of one connected string of conductive yarn, looped together row by row. While this work demonstrated seamless integration of conductive elements into a wearable form factor, it was limited in the layering of conductive elements over each other. Vojtech et al. also developed a fabric-integrated 3-lead ECG system, but they used a prefabricated conductive textile, limiting the design freedom available in fabrication of the electrode [14]. A sewn solution provides more freedom than knitting or preexisting conductive fabrics because of the broad range of patterns that can be used and the ability to overlap stitches to create a denser patch of conductivity within the fabric. Kannaian et al., Pola and Vanhala, Li et al., and Ankhili et al. used sewing and embroidery to integrate silver-coated threads into fabrics. These groups all present promising results, but all except Ankhili et al. lack the sample size to produce statistically significant results. Testing with multiple subjects is important for demonstrating the broad utility of these textile electrodes, independent of interindividual variability.

Conductive ink- and paste-based electrodes are developed by printing a custom-manufactured conductive ink onto the fabric surface. There are a variety of methods employed to adhere the conductive substance to the chosen substrate: dip and dry, sputtering, screen printing, chemical etching, stencil printing, and permeation [9,10,15,16,17,18,19,20]. The method chosen depends on the properties of the conductive substance and the substrate. A challenge with this type of electrode is the response of the conductive ink to fabric strain once it is dry on the surface. Cracking of the ink can happen, causing breaks in the conductive surface and changes in resistance during movement. It is not standard practice to test these electrodes under stretch, bend, and wash conditions (only two of the groups listed in Table 1 conducted wash testing), but it is an important assessment due to the potential for signal-impacting cracks within the electrode.

The advantages of conductive threads and yarns that are knit, sewn, or woven into textiles are the low production cost, ease of manufacture, and design flexibility. Conductive inks and pastes are in general more expensive than conductive threads, and they require curing time to set onto fabrics that conductive threads do not. For these reasons, this research effort is focused on developing smart textile electrodes by integrating conductive elements into textiles using traditional fabrication methods. Here, we present a sewn electrode that can be integrated directly into a textile, connected to garment-integrated wiring, and used for long-term ECG monitoring in the wild.

The first research question we aim to answer is whether a sewn textile electrode can perform ECG monitoring to the same fidelity for the desired metrics as a traditional electrode. Three hypotheses are tested through this comparison: (1) average R-R interval will be the same between both electrodes, (2) average HR will be the same between both electrodes, and (3) average comfort will be the same between both electrodes. While R-R interval and HR both rely on the accurate detection of the R peak, the method of calculation for both metrics differs. While the HR calculation is measured and updated after each beat, R-R interval values cannot be derived from this measurement. R-R interval is the time between beats, which is not recoverable from an average HR value.

The second research question we aim to answer is whether the electrodes’ performance will be impacted by daily use once integrated into a garment. Stretch, bend, and wash testing are proxies for everyday use and are used in this study to test the durability of the electrodes. Our hypothesis is that resistance across the electrodes will not change at different levels of stretch and bend or after multiple wash cycles.

Our third and final research question is focused on the comfort of the wearer—an essential piece to consider in the development of wearable electrodes. We used an existing comfort scale to assess the differences between the adhesive electrodes and sewn electrodes for each subject. No other prior studies has quantified the comparison of signal quality, durability, and comfort between sewn electrodes and adhesive electrodes. Testing with eight subjects allows the rigorous statistical analysis of test results presented here that gives us confidence in our findings.

## 2. Materials and Methods

We developed a set of three sewn electrodes to support 3-lead ECG data collection to compare to traditional electrodes. The traditional electrodes used for comparison consist of a silver/silver-chloride disk mounted on an adhesive patch with conductive gel underneath to enhance skin contact (A10049, VERMED). The silver thread consists of a standard nylon thread coated in silver nanoparticles. This thread is two-stranded with each filament coated separately. The textile electrodes are stitched in overlapping zig zags onto an inextensible fabric backing using a sewing machine (CS5055PRW, Brother), and the conductive area of the electrodes is a 3 × 3 cm square, as shown in Figure 1. Electrode sizing was guided by the work of Yokus and Jur, 2016 [9]. The overlapping wiring paths increase the conductive pathways within the electrode, reducing electrical resistance. The traditional electrodes exhibited a measured resistance of 2.0 Ω, while the resistance of the textile electrodes was 0.3 Ω. Resistance was measured with a standard multimeter. Resistance is a key measurement for textile electrodes because it indicates how easily the electrode will pick up the relatively small voltages across the skin produced by the heart beating. The low resistance of these textile electrodes is essential due to the lack of conductive gel to reduce skin/electrode impedance [21]. A conductive snap was added to the back of each electrode to serve as the connection point between the sewn sensing area and the BIOPAC system connector clips (shown in Figure 1). The snap did not make contact with the skin. Resistance was measured from the clip to the edge of the electrode.

### 2.1. Human Subject Testing

A within-subjects design was selected to reduce the impact of signal variability due to different physiology, sex, and other interindividual differences. A power analysis was performed a priori to determine the number of subjects necessary to see a statistically significant change. To determine differences between traditional and sewn electrodes, a maximum difference between the sensors was set to an observable effect size of 10% of the mean. This mean value was derived from preliminary ECG measurements on a single subject. The power analysis deemed eight subjects necessary to see this effect size. The subject pool consisted of 5 males and 3 females with no history of cardiac illness.

Data collection and analysis for both electrodes was performed using the BIOPAC MP160 ECG data acquisition module (BIOPAC Systems, Inc., Goleta, CA, USA). Electrodes were applied in a three-lead, chest-mounted configuration with one electrode under each clavicle and the third on the lower left rib cage (configuration shown in Figure 2). Each test ran for two minutes with a sampling rate of 2000 Hz. Subjects were seated and instructed to relax and limit motion during each data acquisition period. The traditional electrodes include an adhesive backing for application; the textile electrodes were applied with four pieces of double-sided cello-tape. The textile electrodes required light external pressure to achieve a clean signal. Once fully integrated into a wearable system, this pressure would be applied by a tight-fitting garment. Positioning of the traditional electrodes was marked on the skin and matched with the textile electrodes to control for differences in positioning and their impact on signal quality. Before and after each test, the subject rated the comfort of the electrodes using a comfort survey developed by Hollies et al., 1979 [22]. This survey has been used to assess the comfort of military-grade garments [23]. It consists of 15 words describing sensations that clothing might produce for an individual; the words were selected and validated by their ability to differentiate between two similar garments (tested with a pair of shirts and a pair of jeans) in a test subject population. The subject rated any sensation they felt as partially (4), mildly (3), definitely (2), or totally (1), and if they did not feel a sensation, it was simply left unmarked. Posttest, each word was given a (+) or (−) sign based on the desirable qualities of the electrode. Each ranking was added into a combined comfort score. The scores with the smallest absolute value are the most comfortable. The highest values (i.e., the least comfortable rating for each sensor given by the subject) were used for analysis for each electrode.

The quantitative metrics from the ECG signal were extracted using built-in BIOPAC data analysis packages in the AcqKnowledge 5.0 software. The largest spikes in the ECG signal comprise the QRS complex. This is the strongest part of the signal and the best indicator of each heartbeat. R-R interval is the time between each R peak (the largest positive peak in the signal), and HR is the number of beats per minute (bpm) [6]. From the two-minute sample, 21 QRS complexes were selected for R-R interval calculation. These 21 complexes were selected as the “cleanest” signal, as determined by visual inspection. Each subject had more than enough clean signal to satisfy the requirement of 21 complexes, and variability in signal quality across subjects was minimal. HR was derived from the entire two-minute sample.

### 2.2. Stretch, Bend, and Wash Testing

Wearable sensors are designed with the goal of long-term, everyday use, and the target use case for these textile electrodes is integrating them into a wearable garment. With that use case comes the need to understand changes in the electrodes’ performance during the strains placed on them during natural body movement. Additionally, a garment that is worn day in and day out will absorb sweat from the wearer and collect dirt from the environment, making washability of the electrodes key for long-term use. We conducted stretch, bend, and wash testing of the electrodes mounted onto a swatch of stretchy fabric (jersey knit) to simulate integration into a full garment. We developed an extra set of three textile electrodes to test the effects of stretch and bend testing and wash testing on electrode resistance independently. We chose to test for changes in resistance because this is a key indicator of signal quality during ECG data collection but does not require human subject experimentation.

The electrodes themselves are not stretchy, but they are mounted to stretchy fabric to create tight-fitting garments that maintain skin contact. For this reason, we are testing whether there is any change in resistance of the electrode when the backing fabric is stretched to two different extents and hypothesize that there will be no change. We tested each electrode with 0% stretch (control), 12.5% stretch, and 25% stretch. Stretch testing was conducted on a flat board platform, with the electrode backing clipped to either side. The resistance was tested in four directions across the electrode for each stretch case: vertical (along the stitch direction), horizontal (perpendicular to the stitch direction), corner to corner diagonally, and the opposite corner to corner. Each measurement was repeated three times.

Electrodes integrated into garments may experience bending during use. It is important that this bending does not change the functionality of the electrodes. We conducted bend testing at three different bend angles: 180${}^{\circ }$ (control), 135${}^{\circ }$, and 90${}^{\circ }$. The smallest angle is 90${}^{\circ }$ because we do not anticipate incorporating these electrodes on parts of a garment that will experience bend beyond that angle—areas of the body creating angles smaller than this are not physiologically valuable for ECG monitoring. Each of three electrodes were tested at all three bend angles in the vertical and horizontal directions (parallel or perpendicular to the stitch direction) three times each.

Wash cycle testing included washing and drying phases. We derived a baseline for characterization from our anticipated use case: one month of monitoring with an ECG garment worn four times per week. Assuming average physical activity of the subject, this requires washing the garment twice each week, yielding eight wash cycles for one month of testing. The electrodes were mounted to a swatch of stretchy fabric (jersey knit) to simulate integration into a tight-fitting garment. The electrodes were washed with a “regular”-sized load of other laundry using commercially available liquid detergent with a “normal” wash cycle type. The electrodes were hang-dried over a two-hour period indoors. We aimed to approximate industrial washing standards but could not follow them exactly with the resources available. The resistance across each electrode was measured before the first wash cycle and subsequently after each complete cycle of wash and dry.

### 2.3. Statistical Tests

For the R-R interval and HR data, the first step in data analysis was to test for normality using the Shapiro–Wilk test. Next, the F test for equal variance was performed between traditional and textile samples for R-R interval and HR. Finally, a paired t-test was performed between samples to test the following hypotheses: (1) average R-R interval will be the same between both sets of electrodes and (2) average HR will be the same between both sets of electrodes. For the comfort rating data, the Wilcoxon signed-rank nonparametric test was used to test the last hypothesis: (3) average comfort will be the same between both electrodes. In these analyses, failure to reject the null hypothesis implies there is no statistical difference between the two electrodes in signal quality and comfort.

For stretch and bend testing, comparative statistical tests were not performed due to the small sample size (3 trials for each level of stretch and bend). Instead, descriptive statistics were calculated. Differences between average values for each measurement that lie within the standard deviation of the three trials were considered to be insignificant differences for this purpose.

For wash testing, a correlation test was run between wash cycle and resistance for each of the three electrodes.

## 3. Results

All four sets of ECG-derived data (R-R interval and HR for traditional and textile electrodes) were normally distributed (W = 0.93, 0.99, 0.97, 0.94; *p* > 0.5). Additionally, both sets of data met the requirement of equal variance to pool the variances in a paired t-test (R-R interval: F = 1.32, *p* > 0.5; HR: F = 0.71, *p* > 0.5). The test for difference in R-R interval between the textile and traditional electrodes failed to reject the null hypothesis (t = 1.43, *p* > 0.1). Similarly, the test for difference in HR between the two sets of electrodes failed to reject the null hypothesis (t = −0.70, *p* > 0.5). Table 2 gives the HR values for adhesive and textile electrodes alongside the difference between the two for each subject. The differences in HR are relatively small, and they are not consistently directional.

Table 3 shows the average R-R interval values and their standard deviation for both sets of electrodes on each subject. As with the HR values, the differences between R-R interval values are small. The aforementioned statistical tests support this assessment for both data sets.

Figure 3a,b show the adhesive electrode signal and the sewn electrode signal. The voltage of the adhesive electrode signal is higher than that of the sewn electrode signal, but the key R and S features of the signal are still clearly visible. The lower voltage of the sewn electrode signal is likely due to the capacitive effects of textile electrodes. We hypothesize that the air trapped within the stitches of the electrode causes it to behave as not only a resistor in the circuit but also as a capacitor [8]. The strength of the capacitive effects of the electrodes is not completely understood at this time and is ongoing work. The measurements were not taken simultaneously because we believe the presence of the adhesive electrode impacts the electrical properties of the textile electrode–skin interface. Thus, the data from each electrode are presented separately.

The words of the 15 comfort descriptors selected by the eight subjects are shown in Table 4 along with the average rating of each sensation. The test for difference in comfort ratings also failed to reject the null hypothesis (V = 15, *p* > 0.5).

The four resistance measurements from each electrode during stretch testing were averaged for comparison. The averaged values over the three trials shown in Figure 4 show no significant change in resistance between stretch levels. The change in resistance was less than the standard deviation of the three trials.

Averaged results from three cycles of bend testing (shown in Figure 5) also show no significant change in resistance in the vertical or horizontal directions.

Wash testing showed a slight increase in resistance over eight wash cycles. A correlation test was run for each electrode between resistance value and wash number. This resulted in correlation coefficients of 0.69 (*p* = 0.04), 0.90 (*p* = 0.001), and 0.94 (*p* = 0.0001) for each of the three electrodes. With this increase, the electrodes do not surpass the base resistance measurement of 3 Ω measured from the traditional adhesive electrodes. Resistance values after each wash cycle are shown in Figure 6.

## 4. Discussion

These results show that the sewn electrodes not only produce data clean enough to measure beat-to-beat variability and average HR across eight subjects but also show promising durability after stretch, bend, and wash testing. The ultimate goal of these textile electrodes is broad use, and while many previous studies have conducted human subject testing, it is often performed with only one subject [24,25,26]. This lack of testing on a range of subjects limits the studies’ ability to state electrode functionality across a wide range of anthropometries and skin types. Testing with multiple subjects allows us to analyze the results statistically, removing observation bias from the evaluation of the electrodes and producing statistically powerful results from performance comparisons.

Stretch, bend, and wash testing results are encouraging with no significant changes in resistance within the sensing area during stretch and bend testing and with minimal changes during wash testing. Durability tests are not commonly conducted during the development and evaluation of flexible electrodes. Only a handful of the groups listed in Table 1 conducted wash testing, and only one conducted stretch testing [20]; none of these groups have performed all three evaluations (stretch, bend, and wash testing) together. Ankhili et al. conducted wash testing of embroidered electrodes but only tested changes in resistance within just the conductive lines connecting electrodes to the data acquisition system. Changes in resistance within the electrodes themselves were not evaluated [27]. Testing the resistance of the sensing area after a series of wash cycles is an important contribution to our understanding of the durability of this type of electrode. Kim et al. conducted wash testing on gold-coated fabric and found that the an extra coating over the gold layer was essential to minimizing the impact of washing on the electrodes’ resistance [18]. Electrodes that can be washed can be integrated into a full, washable garment without performance decrements after use. Reusability is key for wearable sensors used in operational environments, so the minimal changes in resistance over eight wash cycles are a positive result to move us closer to a low-burden, fully integrated wearable ECG garment. Motion artifacts are a common problem in textile electrodes, yet few groups evaluate the robustness of the electrode itself under strain and bending, simulating movement required during ambulation. Fabric-based electrodes introduce high opportunity for sensor performance to change due to shifting thread connections, strand breaks, and stretching, so these proxy tests of electrode movement are essential for determining functionality [28]. With many methods of flexible electrode fabrication, bending and stretching can cause small breaks in the conductive substrate (whether it be thread or ink), so it is encouraging to see that resistance of the sewn electrodes is impacted neither by these motions nor by the strain of washing [29]. Our work contributes not only to the design of sewn electrodes that are not impacted significantly by wash, bend, or stretch but also to the accompanying methodology for these tests. Together, this information allows for an understanding of the utility of these kinds of electrodes for repeated, long-term use not found elsewhere in the literature.

Ease and cost of manufacture are other benefits of sewn electrodes in comparison to other flexible electrode counterparts. Conductive inks and pastes tend to be expensive to purchase or manufacture, and the curing process takes more time to be test-ready than sewing [15,16,17,18]. Embroidered electrodes are similar to sewn electrodes because both rely on stitching conductive thread into a fabric for manufacture, but embroidery adds complexity and cost to the process. Ankhili et al. present an embroidered electrode with promising results, but embroidery machines are far more expensive than basic sewing machines, which makes the sewn electrodes a more cost-effective option [27].

Textile electrodes are intended to improve wearability of the electrodes from the user’s perspective, but the user’s perception of the electrodes during testing is often not quantified. Assessments of comfort to the human are not presented by any of the publications listed in Table 1. Despite the fact that we did not see any significant difference in comfort between the adhesive and sewn electrodes, the presentation of a method to assess subject comfort is an important step for the evaluation of flexible electrodes. An electrode that produces a clean signal is only useful if it is comfortable enough to be truly wearable for the common user.

The next step in this work is to integrate the textile electrodes into a garment to test R peak detectability during movement. A challenge for all textile electrodes is maintaining skin contact during movement, as shown in the work of Pola and Vanhala [25]. Sewn electrodes allow a conformal fit that will enable the collection of clean signals, even during movement. The sewn electrodes have characteristic ridges of conductive thread caused by the overlapping of stitches that could provide a solution to this issue without impacting comfort. These ridges help maintain skin contact by pressing into the skin and by preventing the electrode from shifting out of place. The silver-coated thread is soft to the touch, so this enhanced contact will feel like an everyday garment. Additionally, added moisture—such as sweat during use—has the potential to impact signal quality [30]. The design of the sewn electrodes is such that the base fabric can absorb sweat, limiting the added moisture within the conductive part of the electrode and preserving signal quality. We look forward to testing the functionality of the electrodes in these capacities.

## 5. Conclusions

We present the design, development, and test results of sewn electrodes for ECG monitoring. Our results are promising for using this method to integrate electrodes directly into garments for daily use. A truly wearable ECG monitoring system stands to improve the comfort and convenience of not only cardiac health monitoring but also other biopotential signals detected at the surface of the skin such as electrodermal activity (EDA), electromyography (EMG), and electroencephalography (EEG). The sewn textile electrodes presented here reliably record the ECG signal and do not exhibit changes in resistance during stretch, bend, or wash testing. This shows that they are a promising option for implementation into a garment-integrated ECG monitoring system. The testing framework implemented here with eight subjects is not standard in the development of flexible electrodes. The capability of running statistical tests on results removes bias from the analysis of performance, which is important to the accurate presentation of results. With this work, we contribute sewn electrodes that can be integrated into wearable garments for ECG monitoring in operational environments. In addition, the framework we have used to evaluate the durability of these electrodes and the experimental design we executed add a higher level of quantified comparison than we have seen previously in the field of wearable electrode development.

## Figures and Tables

**Figure 1 sensors-20-01013-f001:**
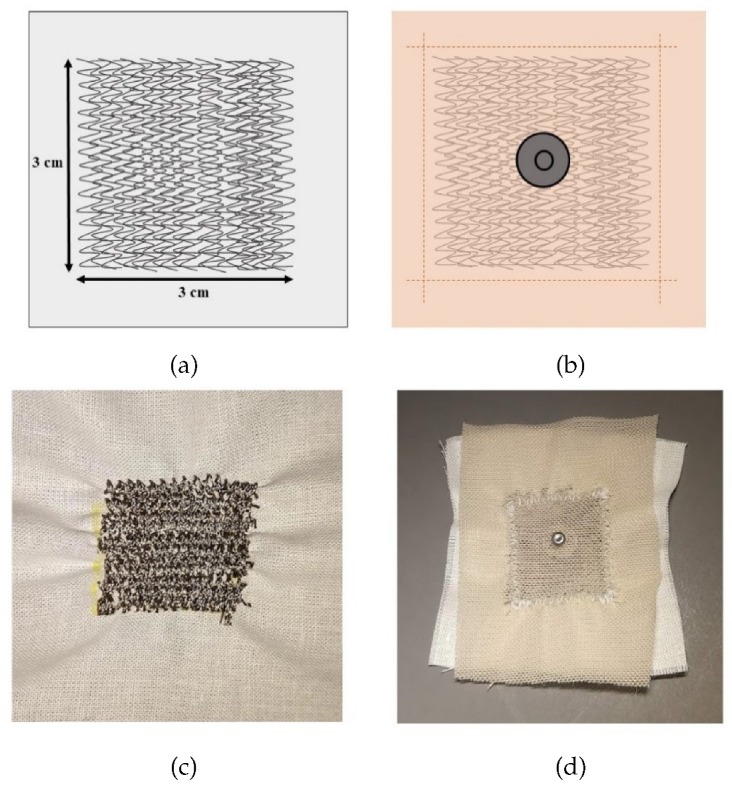
(**a**) Sewn electrode stitch design; (**b**) Design of fabric protective cover and snap connector; (**c**) Stitched electrode; (**d**) Complete electrode with protective cover and snap connector.

**Figure 2 sensors-20-01013-f002:**
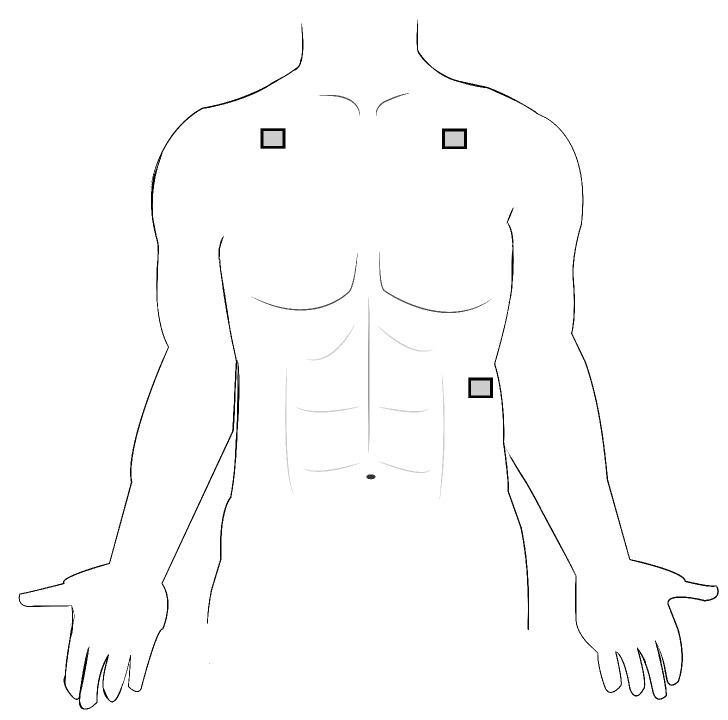
Three-lead positioning for recording of the Electrocardiogram (ECG) signal.

**Figure 3 sensors-20-01013-f003:**
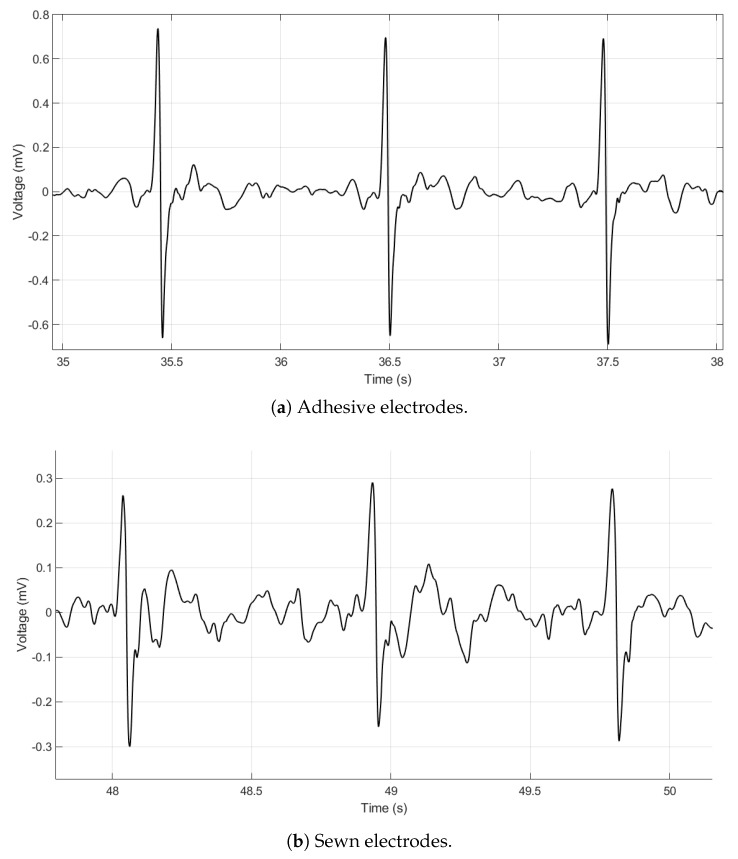
ECG data taken on a seated subject with adhesive electrodes (**a**) and sewn electrodes (**b**). Both signals are filtered with a 60 Hz notch filter and a Daubechies wavelet filter.

**Figure 4 sensors-20-01013-f004:**
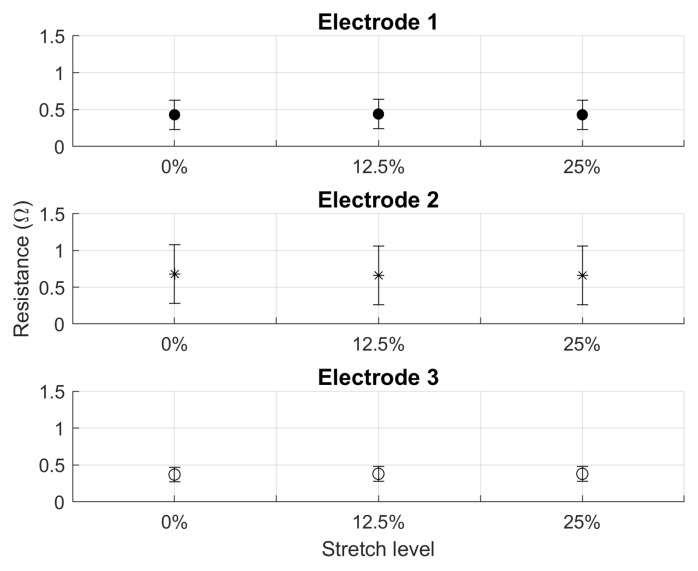
Stretch testing results for each of the three electrodes with error bars representing one standard deviation. Resistance change between stretch levels for all three electrodes is within the standard deviation of the three measurements taken, so there is no observable change.

**Figure 5 sensors-20-01013-f005:**
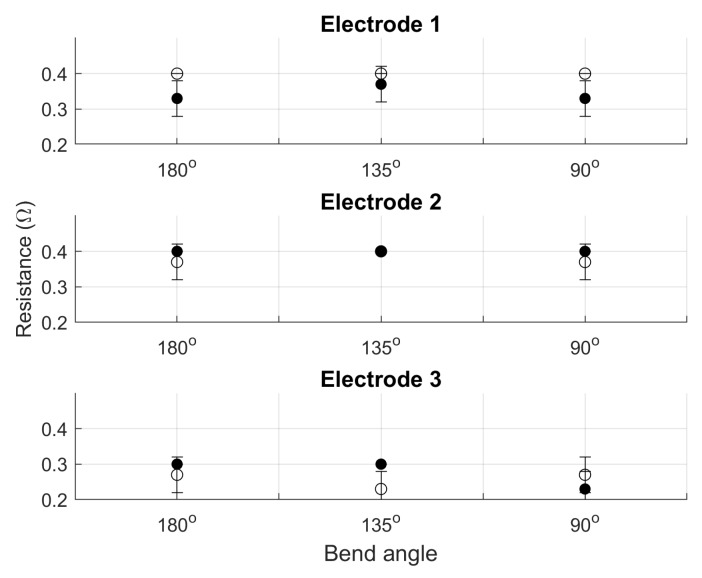
Vertical measurements are the filled circles, and horizontal measurements are the open circles. Many of the measurements returned the exact same results, so not all data points have visible standard deviations.

**Figure 6 sensors-20-01013-f006:**
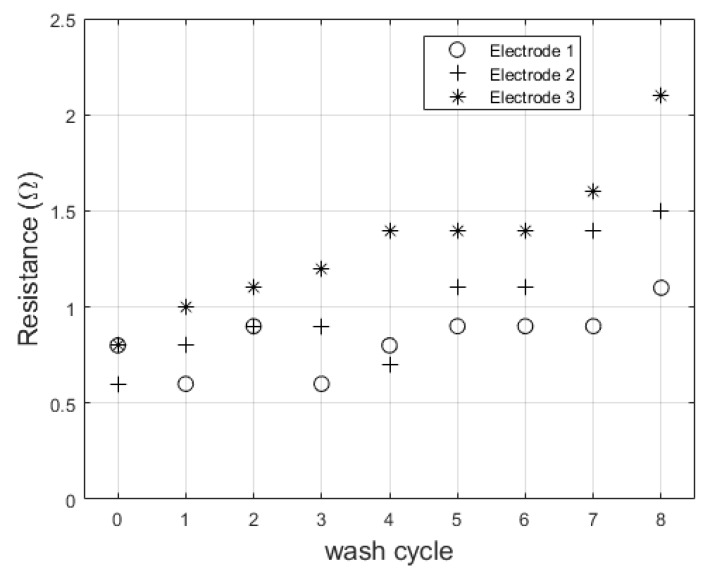
Resistance change after each wash cycle for a separate set of three electrodes.

**Table 1 sensors-20-01013-t001:** This table presents flexible electrode studies found in the literature. Those with an “N/A” value under durability do not list any stretch, bend, or wash testing results. The studies above the divider use conductive threads and yarns, while those below it use conductive pastes, inks, and particles. Note that none of these studies conducted comfort tests.

Group	Conductive Material	Method of Application	Durability Testing
Mestrovic et al., 2007	yarns: silver, copper, steel	knitting	N/A
Li et al., 2020	silver-coated nylon yarn	knitting	N/A
Pola and Vanhala, 2007	silver-coated yarn	knitting, weaving, sewing	N/A
Ankhili et al., 2019	silver-coated thread	embroidery	wash
Kannaian et al., 2012	silver-coated thread	embroidery	wash
Vojtech et al., 2013	pre-fabricated textile	sewing	N/A
Yoo et al., 2009	silver paste	screen printing	N/A
Kim et al., 2009	gold particles	sputtering	wash
Baek et al., 2008	metal particles	chemical etching on elastomer substrate	N/A
Jin et al., 2017	silver ink	stencil printing; dipping	wash
La et al., 2018	silver-particle/fluoropolymer composite ink	permeating into porous textile	stretch
Yokus and Jur, 2016	silver/silver-chloride conductive ink	screen printing	N/A
Lam et al., 2017	graphene ink	dip and dry	N/A
Cho et al., 2011	copper ink	sputtering	N/A

**Table 2 sensors-20-01013-t002:** Heart rate (HR) for textile and adhesive electrodes for each subject.

Subject	Adhesive HR (bpm)	Textile HR (bpm)	Difference (bpm)
1	81	87	+6
2	90	96	+6
3	78	72	−6
4	69	66	−3
5	69	72	+3
6	75	72	−3
7	75	78	+3
8	57	60	+3

**Table 3 sensors-20-01013-t003:** R-R interval for textile and adhesive electrodes for each subject.

Subject	Adhesive R-R int. (s)	Textile R-R int. (s)
1	0.715 ± 0.03	0.686 ± 0.03
2	0.658 ± 0.04	0.633 ± 0.05
3	0.771 ± 0.06	0.833 ± 0.04
4	0.890 ± 0.03	0.894 ± 0.06
5	0.811 ± 0.06	0.758 ± 0.03
6	0.862 ± 0.04	0.863 ± 0.05
7	0.849 ± 0.11	0.798 ± 0.12
8	1.060 ± 0.09	0.995 ± 0.07

**Table 4 sensors-20-01013-t004:** Set of descriptive sensation words selected by subjects and average scores.

Sensation	Snug (−)	Loose (+)	Heavy (−)	Lightweight (+)	Stiff (−)	Sticky (−)
Traditional	1.67	N/A	4.00	1.75	3.50	2.38
Textile	2.43	3.00	3.50	2.00	2.80	2.17
Sensation	**Nonabsorbent (+)**	**Cold (−)**	**Damp (−)**	**Clingy (−)**	**Rough (−)**	**Scratchy (−)**
Traditional	N/A	4.00	N/A	2.00	4.00	4.00
Textile	4.00	4.00	4.00	2.00	2.00	3.00

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
