# Peer review of "Textile Electrocardiogram (ECG) Electrodes for Wearable Health Monitoring"

_sensors, 2020, doi:10.3390/s20041013_

Round 1

Reviewer 1 Report

The authors have revised everything that I commented on.

the authors could check minor things their paper once again before it is published.

Reviewer 2 Report

The author presents findings on the performance of conductive thread sewn into fabric as an ECG electrode. The advantage of such a solution over pre-fabricated conductive fabric patches is the flexibility of the design into garments. Compared to conductive inks, the design is more suitable for standard manufacturing techniques. The work demonstrates that the quality of the ECG signal for both R-R and HR is comparable to wet electrodes.

Although the work was comprehensive, scientifically sound, and explored different aspects such as launderability and resilience to factors like stretch and twist, these finds are not particularly new. Prior work such as by Kannaian (https://doi.org/10.1177/1528083712438069) demonstrates the exact same work using also nylon thread with coated nano silver particles (albeit calling it embroidered vs sewn). This prior work has also explored aspects such as laundering effects. Overall, although this current work is well executed, there does not appear to be enough differentiation with prior work to warrant acceptance as a novel piece of work. If the work is to be considered for publication, the authors need to perform a deeper dive into the literature and draw out prior work that is more comparable to theirs and present what is the improvement upon current literature. 

Reviewer 3 Report

The manuscript, "Textile electrocardiogram (ECG) electrodes for wearable health monitoring" by Arquilla et al., intends to demonstrate the feasibility of ECG monitoring with sewn textile electrodes in a 3-lead, chest-mounted configuration, and compared its result with a traditional gel electrode.  The demonstration is shows enough confidence for ECG use of this sewn electrode.  This paper is a collaborative work between academia and industry, and the goal of the study is patient-oriented, which I believe as valuable. For e-textile researchers, I feel that the most helpful part of this manuscript is the method of human subject testing, statistical tests, and the command of the data.  However, there are a few writing issues that needs to be addressed before the manuscript can be published.

(1) First of all, the novelty of this study must be clearly delineated.  Sewn electrode itself is not a novelty in ECG monitoring (see, for example, Pola, T.; Vanhala, J. Textile electrodes in ECG measurement. In Proceedings of the 2007 3rd International Conference on Intelligent Sensors, Sensor Networks and Information, Melbourne, QLD, Australia, 3–6 December 2007; pp. 635–639).  There have been many other studies after this work within the 13 years.  The current manuscript does not give fair credits to related previous studies, and I feel the novelty part is somewhat glossed over.  However, I agree with the benefit of the current study.  I recommend the authors to give fair credits to previous studies, and clearly state the novelty (or augmented value) that this study provides to the field.

(2) Likewise, recent e-textile works with novel materials (such as nonwoven textiles) to achieve enhanced wearability must also be mentioned.  Two examples may be:  T-G La et al., Advanced Healthcare Materials, 2018, 7, 1801033 and H. Jin et al., Adv. Mater. 2017, 29, 1605848.

(3) In describing methodology, the conducting thread is described simply as "... consists of a standard nylon thread coated in silver nanoparticles."  As the main point of the study is on sewn conducting thread, this is unacceptably neglectful for an academic paper.

(4) About the washing cycle test, there are a few industrial standards to test washing fastness (for example, ISO 105-C06).  If the arbitrary nature of the current study is due to the lack of equipment, I understand that it is not fixable.  However, the existence of the standard and the study's deviation from it must be mentioned.

(5) Patient-comfort-centric approach of the current study is commendable.  I imagine that the questionnaire for Wilcoxon signed-rank non-parametric test must have been coming from previous studies in nursing or in rehabilitation medicine research fields.  If so, references must be cited and the details of the analysis (including the choice of wording) must be discussed in the discussion part.

(6) Compared to Adhesive electrodes, Sewn electrodes show much lower peak voltage and degraded signal-to-noise ratio.  The reasons of these must be discussed.

(7) Conclusion is missing.

Round 2

Reviewer 2 Report

I have reviewed the changes made by the author and noted that the authors aims to position the paper's contribution as testing sewn electrodes with more rigor such that statistical testing can be performed instead of the novelty of sewn electrodes. I thank the authors for that clarification. As the paper stands, I think it is acceptable for publication. 

This manuscript is a resubmission of an earlier submission. The following is a list of the peer review reports and author responses from that submission.

Round 1

Reviewer 1 Report

In this manuscript, authors developed sewn ECG electrodes by using silver-plated thread.

Recently, there are several scientific publications about the sewn electrode for ECG acquisition, such as

How to Connect Conductive Flexible Textile Tracks to Skin Electrocardiography Electrodes and protect them against washing, A. Ankhili, S. u. Zaman, X. Tao, C. Cochrane, V. Koncar and D. Coulon,  in IEEE Sensors Journal. doi: 10.1109/JSEN.2019.2938333

Washable embroidered textile electrodes for long-term electrocardiography monitoring, Amale Ankhili, Shahood uz Zaman, Xuyuan Tao , Cedric Cochrane, Vladan Koncar , David Coulon 2019 / Volume 2 / Issue 3 / Pages 126-135

The novelty of this manuscript is low. And some detail information about the fabrication of electrode is not given. Such as

Page 2, line 68, the conducting silver thread is pure silver or silver-plated thread? The detail information about this yarn is not given.

Page 2, line 72, the method of resistance measurement is not given.

In Fig1, there is a snap button. Is it metallic? If yes, this snap button can be considered as an electrode.

In page 4, line 138, the wash process is not given.

Page 3, line 87, during the ECG signal acquisition process, the state of subject is not well described. They are sitting, standing or running, etc.?

Hence, I do not suggest to accept this manuscript.

Reviewer 2 Report

This papaer showed the possibility of measuring ECG conveniently in daily life using developed textile electrodes.

However, in order for the paper to be published, the following contents must be modified and supplemented.

1)  The authors presented three assumptions as follows. In the three types of hyperports, No. 1 and No. 2 are seemed to be the same assumption because HR is obtained through the R-R interval, if the correct R-R could be measured, then HR could also be calculated correctly.
Rather, it is deemed better for existing and developed electrodes to show the same performance for specific movements of the subjects.

2) The authors explain how and where the developed sensors were attached to the subjects, but it is necessary to use the images or pictures taken to make the reader understand.

3) The measured ECG signal cannot be found. It is obliged to show measured the RAW signals or the filtered signals using the developed sensor compared to the measured signals from the commercial electrodes.

4) The authors say that the ECG was measured using a Biopac, but there is no detailed information on experimental setting. For example, what is the sampling rate for measurement?

5) The authors show only the changed resistance values after stretching, bending and washing the developed sensors in the result section, which also needs to show how the ECG signals have been changed by the changed resistance values.

6) The authors may be able to further improve their paper by referring to the following papers.

  - Adaptive motion artifacts reduction algorithm for ECG signal in textile wearable sensor

  - Wearable textile electrodes for ECG measurement.

  - Performance Evaluation of Textile-Based Electrodes and Motion Sensors for Smart Clothing.

  - Influence of contact pressure and moisture on the signal quality of a newly developed textile ECG sensor shirt.

  - Adaptive motion artifacts reduction using 3-axis accelerometer in e-textile ECG measurement system.

Minor things

1) the specific information are needed for Biopac, for example, Manufacturing contry and company name.

2) Specific information of subjects are needed.

Reviewer 3 Report

The Authors present a well justified and written study on textile based ECG sensors. The methodology and presentation of results are sound. I would recommend the manuscript for publication. 

Round 2

Reviewer 1 Report

Even the authors have made efforts to improve the quality of this manuscript, the novelty is still not enough for a scientific research publication. I suggest to reject this manuscript.

Reviewer 2 Report

The ECG signals shown in Figures 3 and 4 show the results measured at different times. In general, when a proposed sensor is developed and compared to an existing sensor, the performance of the proposed sensor is evaluated by showing the measured signal at the same time and showing the relationship between the two signals.

R-R interval data and HR data measured from eight subjects are not found. It is not enough to show the results of HR and R-R correlations between the proposed sensor and commercial sensor with six lines in the results section. The authors should use tables to display the values of HR and R-R obtained from each subject.